# Resources and Habitat Requirements for Giraffes’ (*Giraffa camelopardalis*) Diet Selection in the Northwestern Kalahari, South Africa

**DOI:** 10.3390/ani13132188

**Published:** 2023-07-03

**Authors:** Francois Deacon, Gert Nicolaas Smit, Andri Grobbelaar

**Affiliations:** Department of Animal Sciences, Faculty of Natural and Agricultural Sciences, University of the Free State, P.O. Box 339, Bloemfontein 9300, South Africaandri.giraffe@gmail.com (A.G.)

**Keywords:** arid environment, diet selection, giraffes, GPS satellite tracking device, Kalahari region, seasons, woody plant species

## Abstract

**Simple Summary:**

To ensure successful introductions and translocations of giraffes, more scientific information is needed on their diet selections, especially concerning the availability of browsing material. To our knowledge, this 15-month study is one of the first to assess giraffe diet selection by fitting global positioning system collars on this species to monitor the spatial distribution and identify the most utilised vegetation and areas. The study population resided in the Khamab Kalahari Nature Reserve in the northwestern Kalahari region of South Africa. The results indicated the clear utilisation of specific plant species of specific areas during different seasons and between different vegetation types and species. This study provides innovative information to understand better how this giraffe population from the northwestern Kalahari region has adapted its diet selections, especially when compared to other giraffe populations from different regions.

**Abstract:**

Diet selection concerning browse availability of giraffes (*Giraffa camelopardalis*) was studied over 15 months in an arid environment in South Africa. A global positioning system collar was fitted to a giraffe individual to assess the specific areas, consisting of different vegetation types, that the population utilised during different seasons. Results are provided on diet selection in relation to browse availability between seasons and vegetation types, including tree densities and the amount of the total evapotranspiration tree equivalents. Diet selections of the giraffe population changed in response to the availability of browse material from July to October. The availability of important resource areas had a significant (*p* < 0.05) effect on the spatial ecology, and an increase in home range size was noted. Information that is important for the well-being of giraffes was identified. This included nutritional stress and the limited variety of the most utilised tree species available for browsing, especially during critical dry periods. The results demonstrate the importance of assessment of giraffes’ diet selection in relation to browse availability, especially before introduction to a new area, to limit the lack of population growth and underperformance. This study provides valuable information towards understanding the resources and habitats required for successful giraffe management.

## 1. Introduction

Limited information is available on giraffe (*Giraffa camelopardalis*) translocations effects on the natural resource and habitat requirements for the diet selection of this large browser species. In most cases, these translocations were undertaken without ecological and/or conservation considerations, such as if the proposed new environment falls within the natural distribution of the species, if the habitat and vegetation are suitable, and whether or not they would have a negative impact on the environment [1]. Although the southern giraffe, *G. c. giraffa* [2], occurs in fairly large numbers in protected areas across southern Africa [2,3,4], specific information on the diet selection and feeding patterns of these animals in arid regions in South Africa is limited [5,6,7,8]. Most provinces have created guidelines for keeping, transporting, and maintaining game species on a reserve and/or ranch [3]. It is recommended by the authors that provinces such as Northern Cape require detailed habitat analyses and suitability reports before being allowed to locate giraffes there.

Very limited evidence is available to confirm whether giraffes naturally and permanently occurred in the Kalahari region of South Africa [9]. Lynch [10] mentions the possibility that giraffes historically occurred in the semi-arid western areas of South Africa, with very few giraffes historically recorded south of the Gariep (Orange) River [11]. With limited information on giraffes’ permanent residency in this environment, we realise and consider that giraffes did migrate and move from one region to another [12]. However, tree phenology and the deciduous nature of the tree species would make it unlikely that giraffes would have been able to stay there permanently without directly affecting the evergreen trees [13].

Historical distributions are not always clear, and with climate and habitat changes from a hundred years ago until now, limited guidelines do not help make informed decisions. Therefore, a national strategy should be implemented for a detailed habitat analysis before future giraffe translocations can be undertaken. Previous assessments on giraffes’ historical distribution have not included the broader Kalahari region in South Africa and bordering countries of Namibia and Botswana’s southern Kalahari [14,15,16,17]. Recently, many giraffes have been introduced to the Kalahari region of Southern Africa for economic and ecotourism purposes onto private land.

Shorrocks and Croft [18] illustrated how little is known about the diet changes of giraffes over annual seasons. A suggested method in comparing diet over seasons is performing physical observations on plant species selected by giraffes and determining how they allocate time throughout the day to feeding activities and food selection [19,20]. The dietary assessment of herbivores is crucial in understanding trophic relationships and providing insight into potential competition with other herbivore species and the influences those herbivores may have on an ecosystem [20,21,22,23,24]. In addition, studies of herbivore diets are useful as they provide the initial step towards understanding the resources and habitat required on which management decisions can be based [21,22]. The specific objectives of this research study were to determine the percentage of daily activities spent on feeding by a giraffe population in the Kalahari region of southern Africa and determine their diet selections for tree species and how these changed seasonally. Many previous studies on the diet selection of giraffes have been performed in areas of their natural distribution [6,19,25,26]. Such information on giraffes’ diet selections, especially in environments where this species historically did not occur naturally, could assist wildlife managers in preventing environmental degradation. The importance of the chemical composition of available browse [6,7,20,22,27,28,29] is acknowledged but did not form part of the scope of the current study.

## 2. Materials and Methods

As previously described by Deacon and Smit [30], this study was conducted in the Khamab Kalahari Nature Reserve (KKNR: −25°48′49.39″ S, 23°25′40.35″ E) situated in a remote area of the savanna Biome in the northwestern Kalahari region of South Africa.

The property is privately owned and registered as a private game reserve. It is fenced with a 2.4 m fence with a total perimeter stretching to a 243 km fence line. As described by Deacon [31], all giraffes were introduced to the area, and at the time of developing the reserve, the owners bought animals from all over South Africa. It was beyond the scope of this study to determine the genetic status and originality of the study population. Considering research by Van Niekerk et al. [28] describing valuable new data on the status of the giraffe population in the Free State and Northern Cape Provinces, a large percentage of giraffes in private ownership in South Africa could be considered hybrids.

The study area was stratified into relatively homogeneous physiographic-physiognomic vegetation units on a 1:5000 scale (see [30]). Homogeneous units formed the basis of this classification method, derived from large-scale aerial photographs or known environmental parameters (geology, topography, climate, soil, etc.). Eleven plant communities were identified, which were further grouped into three major and five sub-plant community types: (1) *Verbena encellioides-Schmidtia kalahariensis* Grassland, (2) *Schmidtia pappophoroides-Vachellia erioloba* Woodland [2.1] *Grewia retinervis-Vachellia erioloba* Woodland, [2.2] *Vachellia erioloba-Senegalia mellifera* Woodland, [2.3] *Eragrostis lehmanniana-Cenchrus ciliaris* Grassland, and (3) *Geigeria ornitiva-Enneapogon desvauxii* Forbland [3.1] *Monechma incanum-Enneapogon desvauxii* Forbland, [3.2] *Eragrostis bicolor-Enneapogon desvauxii* Grassland (see [31]).

The population of giraffes (varying from 2–26) was located and observed throughout 15 months by making use of the location of a global positioning system (GPS) collared female (SAT312Ke) (see [30]). This permitted consecutive data collection on individuals from the population (in close proximity to SAT312Ke) at any time. All observations were made by tracking giraffes by vehicle. Observations were made with binoculars at distances ranging between 20 and 100 m. Giraffe activities were recorded on a voice recorder [32,33] every five minutes from sunrise until sunset each day. Field observations were performed to determine the feeding behaviour of the giraffes using the scan sampling method modified from Altmann [34]. This sampling method entails recording an individual’s activity or state at predetermined intervals when visible. By observing each animal’s activity simultaneously at a given time point, the percentage of time spent on various activities could be calculated [32]. Diet feeding observations were recorded for each plant species utilised at a specific time. Periods when the giraffes’ activities could not be recorded were excluded from the calculations. Recordings were undertaken each month for ten consecutive days for fifteen months, from June 2012 to August 2013. Within these 15 months, seasonal changes were investigated to determine the effect on giraffes’ diet. Four seasons were identified and grouped into summer (wet hot, November–February), autumn (wet cool, March–May), winter (dry cool, June–August), and spring (dry hot, September–October).

After each day, the voice recordings were transcribed and entered into a Microsoft Excel worksheet. A “feeding record” was defined as an individual foraging on one plant (two individuals utilising the same plant were two feeding records, etc.). The frequency of utilisation of each plant species was determined by expressing the number of feeding records per plant as a percentage of the total number of feeding records. Feeding records for each species consumed/hour were totalled and expressed as a percentage of all feeding records for that hour. All observations and data collected were differentiated between the sexes and compared to other studies.

Each dataset recorded for the population’s feeding behaviour and plant species’ consumption was statistically analysed per season with GenStat^®^ [35]. Generalised Linear Mixed Models (GLMM) with non-Normal distributions were used to test for differences between the utilisation of tree species and season effects, as well as the time x season interaction (Fixed-effects). Time was considered as the random effect and was specified per hour. Treatment means were separated using Tukey’s least significant difference (LSD) test at the 5% level of significance [36]. The GLMM analysis was used with the Poisson distribution and logarithmic link to test for differences between feedings [37].

Data were tested using the CANOCO 4 [38] package. Ordination techniques assisted in understanding a local environmental classification by “grouping” the same seasonal information together [39]. Each data matrix (tree species and season) was analysed using the Analysis of Similarities: one-way and two-way crossed to establish whether there were significant seasonal differences (*p* < 0.05) between the feeding selections.

Only the key tree species were further analysed using the data analyst toolbox with Excel, and the rest of the plant species were grouped and are referred to as “other species”. Regression analysis [40] was used to compare which tree species were correlated in terms of utilisation by calculating the correlation coefficient between their percentage utilisation. Tree species were compared, and correlations indicated whether certain species were utilised at the same time and which were utilised at different periods. The correlation matrix showed the value applied to each possible pair of measurement variables.

## 3. Results

### 3.1. Tree Species Most Utilised by Giraffes

A total of 174 plant species belonging to 45 plant families were identified on the reserve. 50% of the species belonged to five families: Poaceae, Fabaceae, Asteraceae, Scrophulariaceae, and Aizoaceae. Table 1 alphabetically lists the 18 (out of 174) woody tree species utilised by giraffes. A limited variety of plant species were utilised by giraffes, of which only one tree was evergreen (*Boscia albitrunca*), with the rest being winter deciduous. In terms of mean frequency of occurrence, the most important species recorded in the diet of giraffes in KKNR were *V. erioloba, Ziziphus mucronata, B. albitrunca,* and *S. mellifera*. Therefore, these four species were considered important resource species in the diet of the giraffes in the study area (Table 1). During this study, important resource species were defined as plant species available for the animals to utilise, and utilisation was 5% or more compared to overall diet preference (Table 1).

During the study period, tree leaf phenology was described, and dry mass was calculated with the BECVOL-model (Biomass Estimates from Canopy Volume) [41] and provided the estimated dry leaf mass (kg D.M. ha^−1^) per tree species. The estimated total per tree species varied seasonally, decreasing the percentage later into the dry season [31]. Variations between the tree species were also observed. The percentage availability of leaves and shoots suddenly decreased from July until October for all tree species. This explains why certain months were considered critical periods, as lower percentages of food were available.

Of 18,999 observations of giraffes feeding, 45% (*n* = 8639) consisted of *S. erioloba*, 20% (*n* = 3758) consisted of *S. mellifera*, 21% (*n* = 3970) consisted of *Z. mucronata,* and 7% (*n* = 1375) consisted of *B. albitrunca* (Table 1). These four woody species combined represent 93% of the total giraffes’ diet in the KKNR. This suggests that the giraffes selected these woody species, although they are not the most abundant species available (Table 1). These four species combined represent 36.2% of all woody species per ha (*V. erioloba* [16%], *S. mellifera* [14.5%], *Z. mucronata* [2.2%], and *B. albitrunca* [3.5%]) (Table 1). The most abundant woody species was the deciduous species, *Grewia flava* (*n* = 306) (40.5% of plants per ha), but only 2% were included in their diet and therefore not considered an important resource species [Figure 1]. However, using the Evapotranspiration Tree Equivalents (ETTEs), these four species combined represent 58.4% of the food provided (*V. erioloba* [26.1%], *S. mellifera* [18.2%], *Z. mucronata* [7.1%], and *B. albitrunca* [7%]) (Table 1). The season had a significant effect (*p* < 0.05) on the giraffes’ feeding on *V. erioloba* and *G. flava* but not for “Other Species” (*n* = 348). *V. erioloba* constitutes almost half of the giraffes’ diet (45%), is utilised throughout the year, and is a key resource species (Table 1).

No significant differences in forage used for “Other species”, such as forbs, herbs, sedges, etc., were recorded. The least prominent tree species the giraffes selected over the entire study period were combined under “other species”. However, when it comes to feeding when “other species” were utilised, 69% of feeding took place at or below 1.0 m (levels one and two), suggesting that giraffes were browsing at lower levels than the preferred levels. The average height of “other species” was 1.0 m, so 69% of browsing was on average-size trees and 31% of browsing was above average height.

### 3.2. Diet Changes over Seasons

Using the GLMM, the Wald statistic (Tukey’s LSD test) demonstrates which of the woody species had a significant effect (*p* < 0.05) on the diet of giraffes (Table 2). Seasonal effects influenced all the prominent tree species utilised by giraffes. *B. albitrunca* was utilised similarly in winter (2.126) and spring (2.053) and lowest in summer (1.348) and autumn (0.923) (Table 2). *B. albitrunca* is a key resource tree species because of the foliage it provides during the winter and spring seasons (Table 1). *S. mellifera* was less utilised during the winter months. The mean utilisation value in summer (3.072) was the highest, followed by spring (2.901), and autumn (2.667), with the lowest in winter (2.115) (Table 2). *S. mellifera* was considered a key tree species because of the foliage it provides during autumn and because it starts re-sprouting earlier than the *Vachellia* tree species in spring (Table 1).

Seasonal utilisation of *Z. mucronata* by giraffes was very low during spring (0.274) and highest during summer (3.035), followed by winter (2.714) and autumn (2.301) (Table 2). Giraffes fed far less on *Z. mucronata* during spring due to the annual absence of leaves (phenology). *Z. mucronata* is still considered a key tree species because of the foliage it provides during winter and autumn (Table 1). Seasonal utilisation of *V. luederitzii* by giraffes was very low during autumn (0.2081) and highest during winter (1.2742), followed by spring (1.2256) and summer (1.0029) (Table 2). Tree leaf phenology again was the reason for giraffes feeding less on *V. luederitzii* during the autumn season (Table 1). Seasonal utilisation of *T. sericea* (*n* = 83) by giraffes was very low in spring (0.0393) and winter (0.0272) and highest in summer (0.5664), followed by autumn (0.1567). (Table 2). Overall, *T. sericea* contributed very little to the diet of giraffes.

Combining the feeding percentages on the three key deciduous species, *V. erioloba*, *Z. mucronata*, and *S. mellifera*, showed that more than 85% of the giraffes’ diet came from these three tree species. Figure 1 illustrates how giraffe diet selections changed over 15 months. *V. erioloba* was very prominent for most months but less so during November and December, *S. mellifera* showed a decrease in the winter months, *B. albitrunca* increased in the winter and spring, and *Z. mucronata* decreased in the spring.

Using ordination techniques and “grouping” the same seasonal information together, it is clear that the diet of giraffes changed seasonally (Figure 2). This change was not necessarily due to preferences but to what was available. If *V. erioloba* were available, it would probably be their main food source throughout the year, but the results show that tree species availability was not constant in different seasons. The ordinations illustrate the same seasonal tendency, having a small overlap between summer and autumn and distinctive groupings based on diet selections in winter and spring (Figure 2). This helps to identify variations in giraffes’ diet over the seasons.

The change in giraffes’ diet from what they utilised most often to what was available for that season is illustrated in Figure 3. The leaf phenology of each woody species illustrates the percentage availability of browse. *B. albitrunca* (evergreen) was utilised as a key resource during the most critical period from mid-winter (July) to the end of spring (October). Spring had only four woody species available for utilisation, with *V. erioloba* (44%) and *S. mellifera* (17%) at their lowest percentage of leaves available and “other species” becoming important during the critical period. Although the leaf phenology percentage of *V. luederitzii* (86%), *G. flava* (90%), and *T. sericea* (90%) were high, these species were less utilised by giraffes overall, and *Z. mucronata* was not available during spring.

### 3.3. Diet Change between Sexes

All observations and data collected were differentiated between sexes and age groups and compared to other studies [31]. How the diet varied over the months is demonstrated when female and male diets are separated (Figure 4 and Figure 5). Males utilised more of the “other species” (Figure 5) from June 2012 to August 2012 compared to the females (Figure 4). Deciduous *Z. mucronata* was less utilised from August to November, and the deciduous *A. mellifera* was less utilised in June and July. *B. albitrunca* was not utilised by males during January and February (Figure 5) but was utilised to a lesser extent by females (Figure 4).

## 4. Discussion

Giraffes have specific daily requirements to survive and maintain their energetics and physical condition [2,43,44]. The three most prominent behavioural factors that these mammals altered to achieve certain requirements [45], and relevant to this study area, included: (1) the choice of habitat; (2) the selection criteria for choosing food items; and (3) the relative proportion of time allocated to feeding and non-feeding activities. Giraffes search for specific tree species, and from the literature, they select growth with more protein-rich and succulent parts, which are vital for their energy balances [11,32]. Different parts of plants, such as shoots, twigs, leaves, thorns, etc., are included in the diet of giraffes [25,46]. Previous studies [5] conducted on the diet selection of giraffes show that they can feed on an unusually large variety of plant species (depending on the environment and availability) and are flexible, shifting their diet selections according to the availability of specific plant species.

The diet of giraffes in a natural habitat was first described by Anne Dag in 1960 [47] by linking the selection to the browse material’s chemistry or the shape of the leaves [11]. Previous behavioural studies indicated that giraffes’ diet could consist of numerous plant species, ranging from 14 to over 100 [5,26,27,32,47,48,49,50,51,52,53,54,55,56]. Feeding behaviour studies conducted on giraffe populations living in arid environments, such as the Kalahari region, include those conducted in Niger [46] and Namibia [7,20], during which high numbers of plant species diversity were recorded.

Their ability to travel long distances may also contribute to their access to a greater variety of vegetation types, as discussed in Deacon and Smit [30]. Due to their height, they can utilise plants beyond the reach of other herbivore species and overcome most of the morphological defences, such as tree structure and thorns, of plants [31,57].

Giraffes are selective feeders that reject dry plant material [48,58,59,60]. If available, giraffes would typically select feeding in areas where *Vachellia* and *Senegalia* (previously known as *Acacia*) species dominate the vegetation [61,62,63,64,65]. They are also classified as concentrate feeders [32,60,66], having small reticulo-rumens concerning body size and requiring more energy per kg body mass to remain active than other ruminants [60]. As such, giraffes will select specific species of plants and highly nutritive plant parts. This allows giraffes to browse frequently throughout the day on plant parts such as leaves, new shoots, pods, and flowers [52,60,67]. Giraffes exhibit a high degree of selectivity when it comes to browsing particular plant species [65,68]. However, their selectivity is only partial when it comes to differentiating between plant parts. The leaves from deciduous plant species formed the dominant component of their diet, but plant parts such as flowers, fruits, and pods were also regularly utilised when available [33,53]. Concerning plant species utilisation, it was found that, similar to this study, giraffes in the Mokolodo Reserve preferred *Acacia* (*Vachellia*) zones and also showed a strong selection of deciduous species such as *Z. mucronata* [5].

A total of 25 plant species were utilised in the study in the Kruger National Park, South Africa [63]. Twenty-eight plant species were listed and browsed in the Willem Pretorius Nature Reserve, South Africa [33]. Apart from the prickly pear (*Opuntia ficus-indica*) and the barley sugar bush (*Pollichia campestris*), all the rest could be classified as woody plant types. Notably, the most intensively browsed species were all thorny, and nearly half were browsed only occasionally during six or fewer months. According to the various criteria applied, *Acacia* (*Vachellia*) *karroo*, *Asparagus laricinus*, and *Z. mucronata* constituted by far the most important components of giraffes’ diet. Together these three species constituted almost 74% of all observations from Theron’s study in the Free State Province [33]. This agreed with findings in the Kruger National Park that 60–70% of the diet of giraffes normally consisted of their most utilised plant types, with 30–40% of the diet formed by 25 to 35 other species [68], which also agreed with the 31 noted plant species from the study conducted in Botswana [5], and also correlates with the findings from the 174 different plant species recorded during this study.

Seasonal variations in the diet of giraffes were seen in the findings of various previous studies [6,25,26,32,33,50,69].

No significant difference (*p* > 0.05) was found in the variety of plant types utilised in the various seasons (wet: 23; dry: 25), and this finding could be ascribed to the general low diversity in plant species and low abundance of the most utilised food plants [33]. As expected, the deciduous *V. karroo* exhibited a higher importance value in the wet season (44%) than during the dry season (33%). As *Asparagus laricinus* retains its leaves until late, their importance value was identical for the various seasons (16%). Apart from *Z. mucronata* being deciduous, the dry leaves and leaves damaged by frost and shoots and available fruits were continually utilised. Therefore, the plant’s relative importance was the same for both seasons, namely 19%, as was the case for *A. laricinus*. The importance value for the rest of the feeding plants combined was 20% for the wet season versus 32% for the dry season. This means that fewer dominant plants were utilised during the dry season, which was also the time when the nutritive value of plants declined [32,70]. We documented that *V. erioloba* pods and *Z. mucronata* fruits were preferred in the autumn. The seasonal pattern in browsing capacity in the current study area corresponded with the seasons and followed the inverse pattern of leaf phenology. For practical and management reasons, this also illustrates that a large area (ha) is needed per browser unit (lower browsing capacity) when leaf carriage is low in July, August, September, and October. During August and September, the months following winter, it was clear that the ha per B.U. (Browser Unit) were higher than in the months after the growing season (April or May).

In the former arid Transvaal in South Africa, it was recorded that *Searsia lancea* was often utilised in the dry season, regardless of the leaves’ strong aroma and bitter taste [25]. *S. lancea* trees in the Wag-‘n-Bietjie Nature Reserve in the central Free State were browsed to such an extent during the dry season that large parts started dying off [33]. Riparian plant species were conspicuously utilised more in the dry season. The quality of the most utilised plant types tended to decrease during the dry season [70]. To maintain a balanced diet, less digestible species were therefore browsed [71]. *Buddleja saligna*, an evergreen shrub seldom browsed during the wet season, was continually utilised by giraffes in the Franklin Nature Reserve in South Africa [48]. Mainly the young shoots on the canopy perimeter of the trees and shrubs are eaten by giraffes [19,20,25,72,73]. Signs of heavy browsing occurred commonly at the end of the dry season, as demonstrated, amongst others, by the high leaf lines of *Salix babylonica* in the Weltevreden Nature Reserve in South Africa.

Similar heavy browsing was also described by other authors [26,50,74,75] when browsing damage in the form of broken branches caused by giraffe bulls was especially evident during the dry season. The breaking of branches allows leaf material to become available to cows and younger individuals, which would otherwise have been out of their reach. Taking into account the short duration the giraffes have been within the reserve, visually, the plant structure and damage were limited, except on the *B. albitrunca,* where you could see a clear browse effect and impact. Giraffe numbers plummeted from 135 individuals to 111 at the time of this study [30,31]. The study area formed part of a nature reserve (KKNR) housing numerous predators and scavengers, such as lion (*Panthera leo*), leopard (*Panthera pardus*), cheetah (*Acinonyx jubatus*), black-backed jackal (*Canis mesomelas*), wild dog (*Lycaon pictus*), brown hyena (*Parahyaena brunnea*), African white-backed vultures (*Gyps africanus*), and lappet-faced vultures (*Torgos tracheliotos*) [31]. The precise causes of deaths could not be determined from the cleanly scavenged giraffe carcasses but it is speculated that, along with predation, a lack in diet quality and/or a combination of unsuccessful adaptation to the new environment, could have added to this decline.

During critical periods and when comparing the results of different studies [32,49,50,54,62,76], giraffes utilised a notably higher percentage of the available trees and shrubs and used herbaceous and alien plant species. Overbrowsing of selected plant species gave rise to abnormal growth forms, leading to lower browsing heights of giraffes [33].

Similar sex differences (Figure 4 and Figure 5) have been documented in Masai giraffe foraging behaviour in Mikumi National Park, Tanzania [77], Tsavo East [78], and on the Athi Plains in Kenya [79]. Feeding ecology and observations in Kruger National Park showed supportive evidence of differences between sexes [63].

## 5. Conclusions

Giraffes rely on key plant species for their main diet, with variations occurring between different populations and habitats. The results from the current study in the Khamab Kalahari Nature Reserve (South Africa) demonstrate that giraffes selected areas with *B. albitrunca* and *Z. mucronata*, but were mostly influenced by the browsing capacity of an area with an abundance of *A. erioloba*. What influences the calculation of browsing capacity for giraffes is the acceptability of the available plant species, the habitat type, the height distribution of the browse material, the phenology of the plant species, and seasonal variations in the availability of browse. The lack of tree species diversity can result in lower dry matter (kg D.M.^−1^) preferred by giraffes, which may directly influence their survival and production. Results indicate browsers will experience difficulty finding foliage material during the critical period from July–October below 2.0 m because the reserve is 20% overstocked. From the results, 1070 more browsers are present than the KKNR fenced area can sustain. Giraffe survival is influenced by the browsing capacity within a fenced-off area due to competition with other browsers for the same resources. The results indicate that browsing capacity should be determined by the quantity of food available during the dry months just before the onset of the new season (August to October), as tree species’ phenology and deciduous nature result in a lack of available foliage. During this time, areas that can be described as “important resource areas”, those which have *B. albitrunca*, may play an important role in the survival of browsers during this critical pre-season dry period. This is one of the first studies to use GPS collars fitted to giraffes to determine their spatial distribution and to identify and monitor their diet selections. During the critical dry months (July–October), most changes in giraffes’ diet occurred, and the greatest differences were between the four important resource species. Ordinations helped to identify tendencies regarding how giraffes’ diets varied over seasons by grouping similar utilisation values. From the ordination data, it was clear there was a seasonal effect on the diet selection of giraffes.

## Figures and Tables

**Figure 1 animals-13-02188-f001:**
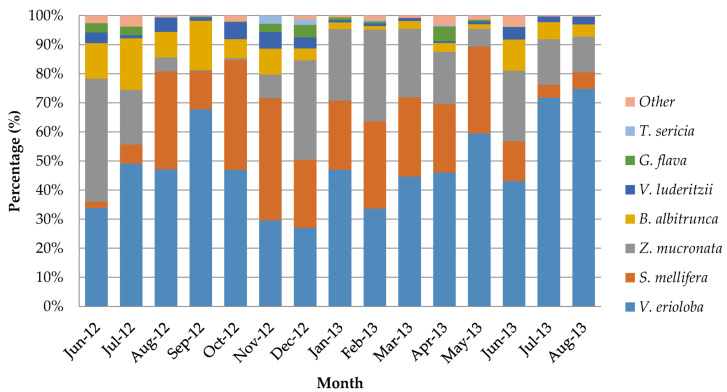
Total percentages of tree species browsed by a population of giraffes over 15 months (June 2012–August 2013) from the Khamab Kalahari Nature Reserve in the northwestern Kalahari region of South Africa (calculated from 18,999 documented feedings).

**Figure 2 animals-13-02188-f002:**
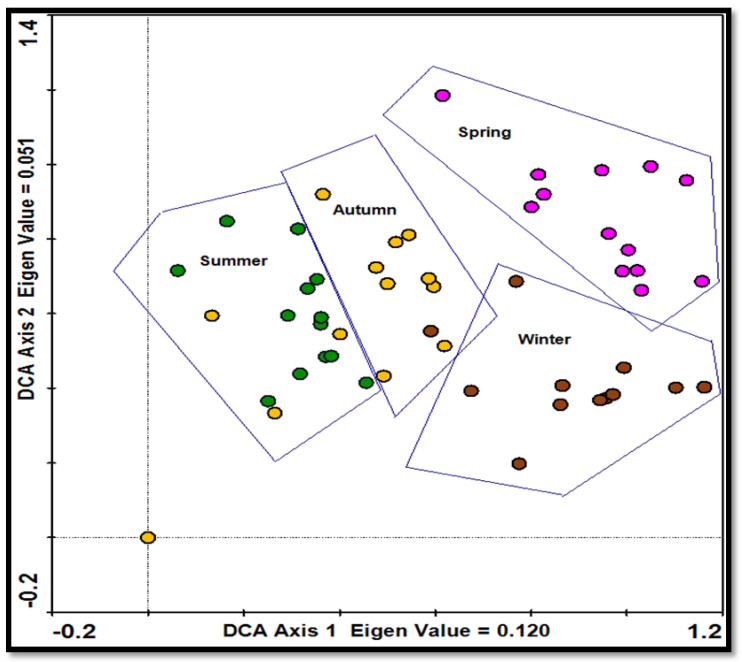
Ordination for giraffes’ diet change over seasons. Diet changes were recorded for a population of giraffes over 15 months (June 2012–August 2013) from the Khamab Kalahari Nature Reserve in the northwestern Kalahari region of South Africa.

**Figure 3 animals-13-02188-f003:**
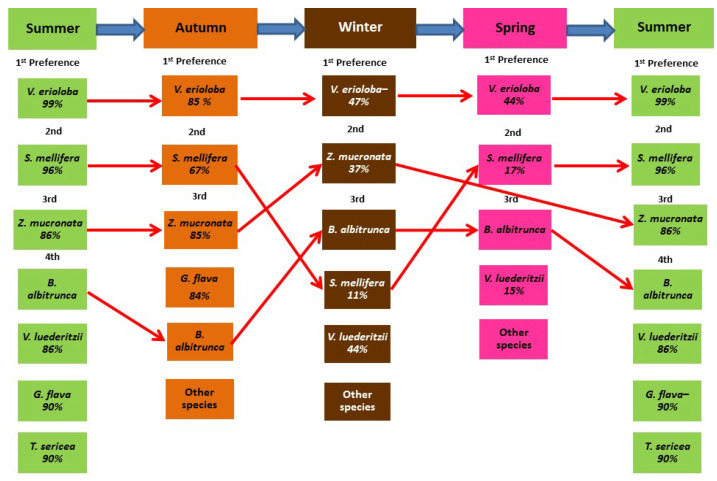
The change in giraffe diet selections over seasons and the percentage of woody species (% of leaves available) in each season (red arrows indicate the trend for each key resource species) for a population of giraffes over 15 months (June 2012–August 2013) from the Khamab Kalahari Nature Reserve in the northwestern Kalahari region of South Africa. Consisting of woody species such as *Vachellia erioloba*, *Senegalia mellifera*, *Ziziphus mucronata*, *Boscia albitrunca*, *Vachellia luederitzii*, *Grewia flava*, *Terminalia sericea*.

**Figure 4 animals-13-02188-f004:**
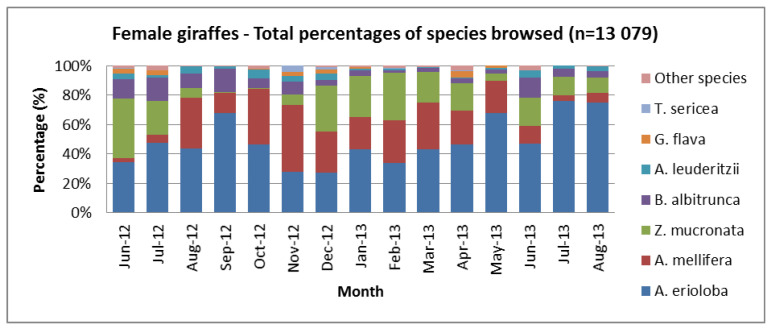
The total percentages of trees browsed by females for a population of giraffes over 15 months (June 2012–August 2013) from the Khamab Kalahari Nature Reserve in the northwestern Kalahari region of South Africa.

**Figure 5 animals-13-02188-f005:**
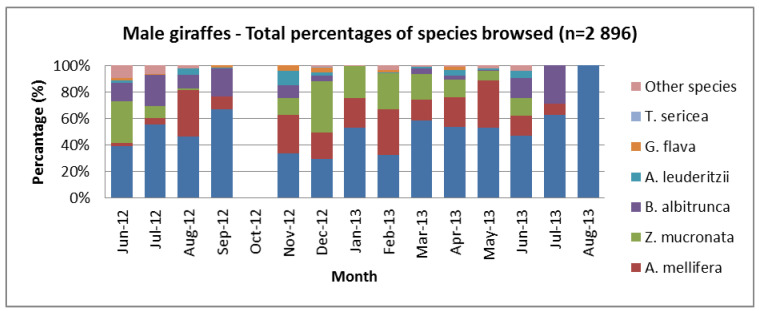
The total percentages of trees browsed by males for a population of giraffes over 15 months (June 2012–August 2013) from the Khamab Kalahari Nature Reserve in the northwestern Kalahari region of South Africa.

**Table 1 animals-13-02188-t001:** Summary of all the woody tree species utilised by a population of giraffes over 15 months (June 2012–August 2013) from the Khamab Kalahari Nature Reserve in the northwestern Kalahari region of South Africa. The plants per hectare (plants ha^−1^) composition and percentage Total Evapotranspiration Tree Equivalent (ETTE) were calculated with the BECVOL model (Biomass Estimates from Canopy Volume) [41,42]. One ETTE = the leaf volume equivalent of a 1.5 m single-stemmed tree.

Species	Important Resource Species (>5% of Diet)	Percentage Plants per Hectare Represented (%)	Percentage ETTE’s Represented (%)
*Boscia albitrunca*	Yes (7%)	3.5	7.0
*Cadaba aphylla*	No	0.1	0.0
*Dichrostachys cinerea*	No	2.4	1.0
*Diospyros lycioides*	No	0.2	0.0
*Ehretia rigida*	No	0.4	0.1
*Grewia flava*	No (2%)	40.5	18.8
*Grewia flavescens*	No	1.3	0.5
*Grewia retinervis*	No	0.6	0.0
*Lycium cinereum*	No	5.2	0.8
*Rhigozum brevispinosum*	No	0.3	0.1
*Searsia tenuinervis*	No	4.9	0.6
*Senegalia mellifera*	Yes (20%)	14.51	18.2
*Terminalia sericea*	No	2.1	5.8
*Vachellia erioloba*	Yes (45%)	16.0	26.1
*Vachellia haematoxylon*	No	0.8	0.2
*Vachellia hebeclada*	No	0.3	0.1
*Vachellia luederitzii*	No (3%)	4.2	13.4
*Ziziphus mucronata*	Yes (21%)	2.2	7.1

**Table 2 animals-13-02188-t002:** Summary of seasonal utilisation (means) of the woody species, which had a significant effect (*p* < 0.05) on the diet of giraffes from the Khamab Kalahari Nature Reserve in the northwestern Kalahari region of South Africa. Test for fixed effects for the giraffes’ diet selections per season (Tukey’s LSD test), listed alphabetically per plant species. Fixed term = season.

Species	Summer	Autumn	Winter	Spring	Wald Statistic	F Statistic	F pr ^1^
*Boscia albitrunca*	1.348	0.923	2.126	2.053	24.67	8.22	***
*Grewia flava*	0.887	0.9865	0.5467	0.1136	14.95	4.98	***
*Senegalia mellifera*	3.072	2.667	2.115	2.901	29.13	9.71	***
*Terminalia sericea*	0.5664	0.1567	0.0272	0.0393	29.61	9.87	***
*Vachellia erioloba*	3.235	3.27	3.573	3.672	8.29	2.76	**
*Vachellia luederitzii*	1.0029	0.2081	1.2742	1.2256	16.4	5.47	***
*Ziziphus mucronata*	3.035	2.301	2.714	0.274	20.25	6.75	***
Other species	-	-	-	-	2.87	0.96	0.414

^1^ *** (<0.001), ** (<0.05).

## Data Availability

The data supporting this study’s findings are available from the corresponding author upon reasonable request.

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
