# Peer review of "Resources and Habitat Requirements for Giraffes’ (Giraffa camelopardalis) Diet Selection in the Northwestern Kalahari, South Africa"

_animals, 2023, doi:10.3390/ani13132188_

Round 1
Reviewer 1 Report
This is an interesting manuscript on the diets of giraffes in the Northwestern Kalahari of South Africa. It describes the main diets of giraffes and compares them to other studies on giraffe diets. The manuscript is well done and easy to follow. However, I have some issues with the introduction and the way it is sold to the reader. Giraffe may not be native to this part of South Africa, but they've been there for quite a while. Saying that this study is needed because we need to know whether giraffe can access adequate foods, and survive seems a bit like a red herring to me. The authors could reword their introduction by simply saying that it would be interesting to see how giraffe have adapted their diets and to what extent the diet there overlaps/or not with that of other giraffes.
This leads me to my second major point, which is about diet preference. Stritly speaking one can not assess diet preference in the absence of a choice. The giraffes only have a choice of what is viable for them there. I would thus suggest avoiding words like "preference" or "avoidance" and using words that describe their choice or use of certain plants instead.
Is there any indication that they do not get enough forage or that there is long-term damage to the vegetation they browse on? Also, the authors mention that they recorded the sex and age of the giraffe but they did not present any age or sex-biased difference in diets, which I think would have been great. Please see the annotate ms for more details.

The quality of English is fine but there are still some grammatical errors and spelling that need to be checked.
Reviewer 2 Report
The MS presented giraffe forage use in an arid population in South Africa. Whilst some aspects of the paper are clear, overall the paper lacks flow, robustness and a limited review of the literature to contexualise the studies findings. The paper concentrates on what is highlighted as preferred forage but essentially looks at observational data without placing it in context of other important factors that influence foraging - chemical composition, energy requirements, preference vs availability, etc.

An extensive English edit is required - see reviewed MS attached highlighting some examples.
Round 2
Reviewer 1 Report
The authors did a great job editing the manuscript and addressing my concerns. I have not identified any major issues. It might be useful, if possible to indicate why the population dropped from 135 to 11 individuals. The causes of death would be useful to know, as the authors claim the reserve might be overstocked. However, in the absence of finding carcasses with clear signs of malnutrition, the statement remains conjecture. Please elaborate on the cause of the decline by listing the cause of death.
Author Response
Thank you very much to Reviewer 1 for a thorough review.
As requested, details on the decline in giraffe numbers have been added.
Updated text include: Giraffe numbers plummeted from 135 individuals to 111 at the time of the study [30,31]. The study area formed part of a nature reserve (KKNR) housing numerous predators and scavengers, such as lion (Panthera leo), leopard (Panthera pardus), cheetah (Acinonyx jubatus), black-backed jackal (Canis mesomelas), wild dog (Lycaon pictus), brown hyena (Parahyaena brunnea) and African white-backed vultures (Gyps africanus) and lappet-faced vultures (Torgos tracheliotos) [31]. The precise causes of deaths could not be determined from the cleanly scavenged giraffe carcasses but it is speculated that, along with predation, a lack in diet quality and/or a combination of unsuccessful adaptation to the new environment, could have added to this decline.
Attached is a document highlighting (via track changes) the changes made to the updated / reviewed document.

Reviewer 2 Report
Appreciate the input the authors have put into reviewing and improving the manuscript. I do feel that it still requires an English edit with some sentences somewhat confusing.
The lack of detailed response from the Authors on the previous review made it a little more challenging to review again, and would recommend in future that they highlight in detail the changes made or not, and present them in a separate document.

See review MS - needs further English edit
Author Response
Thank you very much to Reviewer 2 for a thorough review. All feedback comments were reviewed and suggestions were implemented accordingly. An updated English grammar review was also performed.
Please find attached our responses to the provided pdf document. Please note that a document with track changes (based on the specific edits which were made) will be provided to the Editor.
